# Lossless Image Coding Using Non-MMSE Algorithms to Calculate Linear Prediction Coefficients

**DOI:** 10.3390/e25010156

**Published:** 2023-01-12

**Authors:** Grzegorz Ulacha, Mirosław Łazoryszczak

**Affiliations:** Faculty of Computer Science and Information Technology, West Pomeranian University of Technology in Szczecin, Ul. Żołnierska 49, 71-210 Szczecin, Poland

**Keywords:** lossless image coding, linear prediction, entropy coding

## Abstract

This paper presents a lossless image compression method with a fast decoding time and flexible adjustment of coder parameters affecting its implementation complexity. A comparison of several approaches for computing non-MMSE prediction coefficients with different levels of complexity was made. The data modeling stage of the proposed codec was based on linear (calculated by the non-MMSE method) and non-linear (complemented by a context-dependent constant component removal block) predictions. Prediction error coding uses a two-stage compression: an adaptive Golomb code and a binary arithmetic code. The proposed solution results in 30% shorter decoding times and a lower bit average than competing solutions (by 7.9% relative to the popular JPEG-LS codec).

## 1. Introduction

At present, the processing of images and video sequences has a fully digital structure, and high memory requirements related to storing multimedia data are therefore a significant problem. Reducing memory requirements is possible due to lossy and lossless compression, and the latter type is the subject of this work. The main applications of lossless images and video compression include archiving 2D, 3D, and 4D medical images (three-dimensional video sequences) [1,2,3,4,5] and astronomical images, as well as satellite image compression [6,7]. In addition, the lossless mode is often required during the graphic processing of photos or in advertising materials in the production of television broadcasts and film post-production [8], etc. In the case of the lossy compression of a video sequence, where the division into a group of pictures (GoP) is introduced, the first frame (type I) is coded in an intra-frame mode, whereas the remaining $\left(N_{GoP}-1\right)$ frames are coded in an inter-frame mode. The image is divided into small squares in such types of codecs (similar to the JPEG format used in digital cameras). Further encoding is usually based on the DCT transform. In the type I frame within each square, the DC coefficient, as the first of the DCT-transformed components, determines the arithmetic average of the pixel values in each square. This means that the diminished version of the selected frame is used, coded with lossless compression methods. For example, for a video sequence of 4K quality (3840 × 2160 resolution) and squares of 8 × 8 pixels, we obtain a diminished resolution frame of 480 × 270.

Two steps are usually used in modern compression methods: data decomposition and compression by efficient entropy methods, the most effective being arithmetic coding and Huffman coding [9]. The decomposition is designed to significantly reduce data redundancy resulting from the high level of correlation between neighboring pixels. At this stage, wavelet methods (e.g., JPEG2000 [10], SPIHT [11], ICER [12]) as well as predictive methods (PRDC [13], JPEG-LS [14], CALIC [15]) are used. The most common definition of nonlinear predictors used for lossless image coding is that the prediction function is a linear combination of nearest-neighbor pixel values using a switching scheme between several prediction models associated with particular contexts. Such nonlinear predictors are sometimes called multichannel predictive coders [16]. Two such proposals will be discussed in Section 3.

In contrast, for example, predictive models based on neural networks are characterized by full nonlinearity. In practical applications, an adaptive method is used in which the network weights are modified on the fly after each successive pixel is encoded (methods with backward adaptation). The papers [17,18,19] used an adaptive neural network (AdNN), the design of which was based on a Multilayer Perceptron Network (MLP), whereas the paper [20] used a cellular neural network (CNN). The feature of backward adaptability refers to the time symmetry of encoding and decoding. In cases where neural networks are used, it usually means long process times for both encoding and decoding. When neural networks with forward adaptation (pre-learned using an image database) and lossy-to-lossless coding mode are used, especially when using the latest GPU parallelization and acceleration technologies, the problem of excessive complexity does not exist.

Among the methods with high implementation complexity, there are other time-symmetric solutions with higher efficiency than neural networks. In such cases, the methods were based on linear prediction models with backward adaptation, using mechanisms known from the literature such as RLS [21], OLS [22,23,24], or WLS [25,26], where the coding of each successive pixel is also accompanied by a procedure for adapting or redetermining the coefficients of the linear predictor. The cascaded combination of predictors allows us to obtain the highest compression efficiency, as shown in [25], but it is associated with too long a decoding time (see Section 4). One-step coding (without the step pre-transforming the data) has similarly long encoding and decoding times. [27].

Therefore, this work focuses on time-asymmetric methods (with forward adaptation) with a fast decoding mechanism since the encoding operation is most often performed once, and the decoding operation is performed many times. Backward-adaptive methods for determining prediction coefficients only have pixels already available as decoded on the decoder side (details in Section 2.1), hence learning the prediction model on the fly on both the encoding and decoding sides. In contrast, in methods with forward adaptation, we can already more precisely "tune” the prediction model at the encoding stage thanks to complete information about all image pixels. The paper will discuss different approaches to determining prediction coefficients, including comparing the classical method based on minimizing the mean-square error and the proprietary algorithms presented in Section 2.2 and Section 3.2, which use non-MMSE solutions with different levels of implementation complexity at the coding stage. Section 2.2 discusses the advantages and disadvantages of approaches with minimized prediction errors based on the generalized Minkowski distance. Section 2.3 presents the fast prediction methods using context switching, which are then used in the final solution discussed in Section 3, whereas maintaining the low implementation complexity of the decoder, the proposed solution offers a 7.9% lower bit average relative to the popular JPEG-LS codec. In Section 4, we present a comparison of the effectiveness of the proposed solution with competing methods of similar complexity (methods with forward adaptation). The decoding time of the proposed solution is 30% shorter than the relatively fast Multi-ctx method [28].

## 2. Application of Linear and Nonlinear Prediction in Lossless Image Coding

### 2.1. Practical Aspects of Lossless Image Coding

Data modeling for two-dimensional signals such as images boils down to removing as much mutual information between adjacent pixels (spatial correlation) as possible. To do this, the predicted value (rounded to an integer value) of the encoded pixel ${\hat{x}}_{n}$ is determined. Then, the differences between them are encoded, which are referred to as prediction errors, which are most often small values oscillating near zero: (1)$$e_{n}=x_{n}-\left[{\hat{x}}_{n}\right].$$

In this way (i.e., by encoding successive rows of the image from top to bottom and within a row and consecutive pixels from left to right, which leads to a linear complexity of decoding time as a function of the number of image pixels), we obtain a differential image in which the errors distribution $e_{n}$ is close to the Laplace distribution, which allows us to encode them efficiently using one of the entropy methods. With an eight-bit input data scale, the prediction error value is an integer in the interval $e_{n}\in \bar{-255;255}$. One proposal for determining the predicted value is to use linear prediction with an appropriate selection of neighboring pixels and the prediction order. To obtain a one-dimensional data vector from the neighborhood, it is necessary to number the neighbors of the encoded pixel, for example, according to the increasing Euclidean distance $\sqrt{{\left(\Delta x_{j}\right)}^{2}+{\left(\Delta y_{j}\right)}^{2}}$ of their centers. The numbering of pixels with the same distance is then determined clockwise. This allows the apparent one-dimensional domain of the signal, which facilitates the mathematical notation of many formulas and the use of known formulas themselves, concerning one-dimensional signals. We discussed the analysis of other numbering methods in paper [25]. Figure 1 illustrates the 46 numbered nearest neighbors of a coded pixel $x_{n}$, where the given *j*-th number points to a pixel with a value P(j).

The linear predictor of order *r* has the form: (2)$${\hat{x}}_{n}=\sum_{j=1}^{r}b_{j}\cdot P\left(j\right),$$
where the elements $b_{j}$ are the prediction coefficients that make up the B vector to be passed in the file header to the decoder [9]. It is widely accepted that, for images, the determination of prediction coefficients by minimizing the mean squared error (MMSE) gives very good results [9,25]. To ensure that the header size not too large and assuming that the prediction coefficient is between −1.999 and 1.999, we can write each coefficient using a small number of bits, that is, (N+2) bits. Each bit is used to store the sign of a number and an integer value, respectively. The remaining *N* bits are the precision of the fractional part. To achieve this, the following operation is performed: (3)$$\bar{B}=\left\lfloor B\cdot 2^{N}+0.5\right\rfloor \cdot 2^{-N}.$$

It is assumed that for images, the sum of the coefficients is equal to 1. Suppose the sum of the prediction coefficients (for example, determined by the MMSE method) is slightly different from 1 (which can be affected by the rounding used in formula (Equation 3)). In that case, we can modify the first coefficient $b_{1}$ by adding a value of $1-\sum_{j=1}^{r}b_{j}$. When using a single *r*-order predictor, the cost of the header is only (r−1)·(N+2) bits. For example, for an image with a resolution of 512 × 512 pixels, with r=24 and N=12, the cost of the header is only $L_{head}=0.00123$ bits per pixel. Therefore, in this work, we can make some simplifications, such as omitting the header part from the formula for the bit average $L_{avg}$ and using the entropy function: $L_{avg}=L_{err}+L_{head}\approx L_{err}\approx H$, where *H* denotes the first-order entropy value of the differential image data (after predictive coding):(4)$$H=-\sum_{i=e_{min}}^{e_{max}}p_{i}\cdot log_{2}p_{i},$$
where $p_{i}$ is the probability of occurrence of a symbol from the prediction error alphabet with index *i*; in this case, the error alphabet can be considered a set of integers with values between $e_{min}$ and $e_{max}$. With this assumption, the MDL (Minimum Description Length) algorithm, which minimizes the bit average $L_{avg}$, is reduced to minimize the entropy function of the differential image data.

### 2.2. Determination of Prediction Coefficients Using the Non-MSE Method

In determining linear prediction coefficients, the MMSE criterion [9] is not synonymous with obtaining a model that allows the smallest possible value of entropy *H*, nor the smallest bit average $L_{avg}$ received after considering header data, for example, adaptive arithmetic coding of prediction errors. The paper [29] proposed a minimization of the mean absolute error (MMAE), which, when the image was divided into squares of size 8 × 8, yielded better results compared to the use of MMSE. A broader justification for the suboptimal effect of the MMSE on the entropy value was presented in the work of [30]. In the paper [25], we showed that depending on the prediction scheme used (cascade prediction) and the size of the learning area *Q*, it makes sense to use different values of the power of *M* (from 0.6 to 2) in the minimization criterion $L_{M}$, which uses the generalized Minkowski distance formula (for M=2 we obtain the Euclidean distance (MSE), and for M=1, the Manhattan distance (MAE)): (5)$$L_{M}={\left(\frac{1}{Size(Q)}\sum_{n\in Q}{\left|e_{n}\right|}^{M}\right)}^{\frac{1}{M}}$$
For methods with iterative optimization of the prediction model, it is more convenient to use the substitute of the objective function $L_{M}$ in place of the bit average $L_{avg}$, even when the simplified assumption of $L_{avg}\approx H$ is made. In the approach proposed in this work, the area *Q* is the entire differential image, for which good compression efficiency is obtained at M=0.75.

Figure 2 shows the dependence of entropy as a function of the parameter *M*; for example, the Noisesquare image was encoded using a linear predictor of order r=8.

Beyond the integer values M=1 and M=2 (especially with M<1 and $b_{j}$ coefficients ranging from −1.999 to 1.999), finding an optimal linear predictor is challenging. In the paper [31], we presented a suboptimal algorithm for determining prediction coefficients using a selective search (referred to here as the Iterative Search Algorithm, ISA), which allows us to minimize the value of Formula (Equation 5) simplified to the form: (6)$${\bar{L}}_{M}=\sum_{n\in Q}{\left|x_{n}-{\hat{x}}_{n}\right|}^{M},$$
where the predicted value is determined from Equation (Equation 2) with the limitations above imposed on the range of coefficients $b_{j}$.

Figure 3 shows the comparison of average entropy values obtained using the ISA algorithm (for a base of 45 test images) as a function of the order of prediction obtained using the four following minimization criteria: MMSE (M=2)—points marked with a diamond; MMAE (M=1)—points marked with a square; Minkowski (M=0.75)—points marked with a triangle; and the entropy function *H*—points marked with a circle. Both Figure 2 and Figure 3 show how distant the goal of entropy minimization is from the results obtained by classical mean square error minimization.

One of the advantages of the ISA algorithm is the flexibility of using an internal (substitute) objective function. Although in the case of compression, we are usually concerned with minimizing the entropy function in the final coding step, deciding to modify the prediction coefficients in subsequent iterations based on the entropy drop does not guarantee that the smallest entropy value will be obtained at the end of the selective search. Comparing the results of using two different internal target functions, in about half of the cases, using the function (Equation 6) at M=0.75 gave a slightly better result than using the function (Equation 4). Our target proposal is an experimentally selected compromise that combines both of these criteria into a single internal (substitute) target function: (7)$${\bar{L}}_{Opt}=H+\sum_{n\in Q}{\left|x_{n}-{\hat{x}}_{n}\right|}^{0.6}.$$

This is due, among other things, to the fact that the ISA quickly takes into account the effects of an additional block of removal of the context-dependent constant component when minimizing errors, which improves the predicted value and which is assumed by the target solution scheme of the encoder proposed here (see Section 3.1).

The ISA algorithm introduces the condition $\sum_{j=1}^{r}b_{j}=1$ and a predetermined precision for writing coefficients to *N* fractional bits. First, we set the original vector of coefficients as B=[1,0,…,0] and the objective function that will be minimized. Either Formulas (Equation 4), (Equation 6), or (Equation 7) can be used. Each *t*-th iteration consists of checking the objective function after encoding the image using a predictor B to which a specific scaled modifier vector ΔB has been added: (8)$$B\left(t\right)=B\left(t-1\right)+2^{-i}\cdot \Delta B,$$

If a reduction in the value of the objective function is obtained, then this vector is stored as the new value of B. For the sum of the prediction coefficients to remain constant, the sum of the elements in the modification vector ΔB must be equal to 0. A well-performing restriction of the set of $\Delta B=[\Delta b_{1},\Delta b_{2},\ldots ,\Delta b_{r}]$ vectors to those with two non-zero $\left\{-1,1\right\}$ elements, whereas the remaining $\Delta b_{j}$ elements are 0, was adopted. The number of modifying vectors is $N_{\Delta B}=r\cdot \left(r-1\right)$.

The ISA involves selective iterative searching, in which only a rough fit of the predictor to the data is obtained initially. Only subsequent steps allow each prediction coefficient to be refined with increasingly smaller bits after the decimal point, so the *i* parameter in Formula (Equation 8) is changed sequentially from 0 to *N*. For each fixed value of *i*, we perform encoding and reading of the objective function for successive vectors ΔB. Going through the cycle of the entire set of $N_{\Delta B}$ modifying vectors, the process is repeated if there is a finding of at least one better vector B.

Usually, there are several such cycles, and we can limit the number of cycles to two for i<4 and to eight for larger values of *i*. *i* denotes the number of test image encodings $N_{ISA}\leq \left(4\cdot 2+\left(N-3\right)\cdot 8\right)\cdot r\cdot \left(r-1\right)=8\cdot \left(N-2\right)\cdot r\cdot \left(r-1\right)$, which is an upper limit, although, in practice, about half of this value is sufficient. Test coding should be a learning process (improving prediction efficiency in subsequent iterations). Thus, the second step related to proper prediction error compression is not performed at this stage. For this reason, the coding time is negligible compared to the preliminary stage of determining the predictive model. Only using the final prediction model leads to an entropy coding procedure of prediction errors.

For example, with N=7,r=14, we have a maximum number of 7280 test image encodings. However, based on experiments for 45 test images, the actual number of executions of the encoding procedure was less and averaged 3746. This is still low in complexity compared to a brute force complete search method that checks every possible state of vector B under imposed constraints. In this case, we are dealing with exponential complexity since the maximum number of test encodings is $N_{BF}=2^{\left(N+2)\cdot (r-1\right)}$. Table 1 presents the value of $N_{BF}$ and the reduced number (after considering the $\sum_{j=1}^{r}b_{j}=1$ condition) for small N=5 and r≤5 values.

In the 27 experiments performed (using the nine test images from Table 1 sequentially for r=2,3,4), only in one case did ISA obtain a slightly higher (increase of fewer than 0.001 bits/pixel) entropy value relative to the entire search. It should be noted that for higher orders of prediction, such high efficiency of suboptimal ISA is not preserved. However, the main advantage of using this algorithm is obtaining good results with a low number of fractional bits (N<7) used to store prediction coefficients, as confirmed by the results of an experiment comparing the average entropy value (for a base of 45 test images at r=14) obtained by the two methods (Figure 4). In the first case (dashed line), the classical minimization of the mean square error was used. Then, the accuracy of the prediction coefficients was reduced using Formula (Equation 3), and in the second case (solid line), the ISA implementation was used with the goal minimization function (Equation 6) at M=2. On the other hand, the disadvantages of ISA include the low efficiency of improving entropy minimization for values N>7 and the long time to determine prediction coefficients for large orders.

### 2.3. Prediction with Context Switching

Although most linear prediction methods with a backward adaptation of the predictive model have relatively high implementation complexity, there are other fast methods that use context-switching of the fixed predictive model. The context is a set of features that characterize the type of the closest neighborhood of the encoded pixel. Using contextual partitioning by detecting different types of neighborhoods (classes with distinct characteristics), we can individualize predictive models well-matched to neighborhood characteristics, increasing compression efficiency. In the most straightforward solutions, the predictors associated with a given context are fixed, and there is no need to determine them separately for each image. An example is the median adaptive predictor (MAP) proposed in the JPEG-LS algorithm based on the median edge detector (MED) context switching technique [14,32]. Several developments of this idea have emerged, including MED${}_{+}$, presented in the paper [33]. Still, the resulting improvement was insignificant compared to methods with more context, the principles of which will be shown in the following sections.

#### 2.3.1. Gradient-Adjusted Predictor

The context-dependent prediction method with fixed coefficients of each model has been proposed as a primary prediction method in the CALIC algorithm [15]. It uses several neighboring pixels to determine the predicted value and allows the selection of one of seven contexts. The decision to choose the appropriate context is made based on the number $d_{GAP}=d_{h}-d_{v}$, where $d_{h}$ and $d_{v}$ are defined as the levels of the neighborhood gradients [15]
(9)$$\begin{array}{c} d_{h}=\left|P\left(1\right)-P\left(5\right)|+|P\left(2\right)-P\left(3\right)|+|P\left(4\right)-P\left(2\right)\right|\\ d_{v}=|P\left(1\right)-P\left(3\right)|+|P\left(2\right)-P\left(6\right)|+|P\left(4\right)-P\left(9\right)|. \end{array}$$

The range of $d_{GAP}$ values is divided into seven fields based on three threshold values $T_{1}=8$, $T_{2}=32$, $T_{3}=80$ (research conducted in the paper [34] suggests changing these thresholds to $T_{1}=6$, $T_{2}=25$, $T_{3}=78$). The following is presented as a pseudo-code Algorithm 1 for determining the *k*-th context number [15].
**Algorithm 1** Algorithm for determining the context number1: **if** 
$d_{GAP}>T_{3}$ 
**then**2:  k←73: **else if** 
$d_{GAP}<-T_{3}$ 
**then**4:  k←65: **else**6:  k←17:  **else** $d_{GAP}>T_{2}$ **then**8:   k←59:  **else if** $d_{GAP}>T_{1}$ **then**10:   k←411:  **else if** $d_{GAP}<-T_{2}$ **then**12:   k←313:  **else if** $d_{GAP}<-T_{1}$ **then**14:   k←215:  **end if**16: **end if**

A modified version of the method with slightly higher efficiency, denoted hereafter as GAP${}_{+}$, was presented in the paper [35]. Each of the seven contexts was assigned an individual fixed linear predictor using two to five of the six neighboring pixels. Table 2 shows the prediction coefficients for each of the seven contexts.

#### 2.3.2. Prediction Method with Gradient Weights

The gradient-based selection and weighting pixel predictor (GBSW) method presented in the paper [36] is based on directional gradients determined similarly to the GAP${}_{+}$ method. After some of our improvements, four values are determined: (10)$$\begin{array}{cc} d_{w}= & \left(2|P(1)-P(5)|+2|P(2)-P(3)|+2|P(3)-P(7)|+2|P(2)-P(4)\right|\\ \; & +|P(6)-P(8)|+|P(6)-P(9)|)/10\\ d_{n}= & \left(2|P(6)-P(2)|+2|P(1)-P(3)|+2|P(3)-P(8)|+2|P(4)-P(9)\right|\\ \; & +|P(5)-P(7)|+|P(7)-P(11)|)/10\\ d_{nw}= & (2|P(1)-P(7)|+2|P(2)-P(8)|+|P(3)-P(11)|+|P(4)-P(6)|)/6\\ d_{ne}= & (2|P(5)-P(3)|+2|P(2)-P(9)|+|P(1)-P(2)|+|P(3)-P(6)|)/6 \end{array}.$$

Predictors P(1), P(2), P(3), and P(4) are associated with these values, respectively. Then, the two with the smallest values are determined from among these four gradients, which become the weighting coefficients of the predictive model. This is a linear combination of two of the four nearest neighbors, with the weights associated with the predictors via the cross method. For example, if the two smallest values are $d_{w}$ and $d_{n}$, the prediction value is determined as follows: (11)$${\hat{x}}_{n}=\frac{d_{w}\cdot P\left(2\right)+d_{n}\cdot P\left(1\right)}{d_{w}+d_{n}}.$$

We developed this method into GBSW${}_{+}$ by adding a fifth gradient $d_{GAP}$, the arithmetic mean of the four described by Formula (Equation 10). The fifth associated predictor is determined as the value of predictor GAP${}_{+}$. When the denominator in Formula (Equation 11) is 0, the predicted value is obtained from model GAP${}_{+}$.

#### 2.3.3. Prediction Method with Multi-Context Switching

In addition to the previously mentioned methods, new proposals continue to emerge with varying numbers of contexts [16,37,38,39,40,41,42]. As a generalization of this concept, it is possible to design a fast function that determines a much larger number of contexts (e.g., 2048). For each, an individual predictor is calculated based on a learning image database. In the paper [28], we proposed this type of solution (Multi-ctx) with five different context determination functions built in parallel.

Thus, the predicted value of the encoded pixel $x_{n}$ was determined as a weighted average using five linear predictors of order 12 and the two fast predictors mentioned earlier with GAP${}_{+}$ and MED${}_{+}$ context switching: (12)$${\hat{x}}_{n}=\frac{1-2\beta }{5}\cdot \sum_{j=1}^{5}\sum_{i=1}^{12}b_{\left(j,i\right)}\cdot P\left(i\right)+\beta \cdot \left({\hat{x}}_{GAP_{+}}+{\hat{x}}_{MED_{+}}\right),$$
with the value of β=0.025 is chosen experimentally. After considering the adaptive arithmetic encoder (described in the paper [21]), both the Multi-ctx encoder and decoder offer relatively low complexity, similar to the CALIC method. Lennagrey’s image decoding time (512 × 512 pixels, using a 3.4 GHz i5 processor and an unoptimized version of Multi-ctx code) is 0.35 s.

Table 3 presents a comparison of entropy values for several fast prediction methods.

## 3. Scheme of the Proposed Solution

The reference point for designing the new solution is the aforementioned Multi-ctx method. Section 3.1 presents the proposed improvements over Multi-ctx, resulting in a reduction in decoding time. In contrast, Section 3.2 discusses the proposal of a fast algorithm for determining prediction coefficients using the non-MMSE method, which offers some trade-off between speed of operation and bit-average level.

### 3.1. Decoder Complexity Reduction

Obtaining further acceleration of the Multi-ctx decoder requires simplifying the Formula (Equation 12). In place of the MED${}_{+}$ predictor, we propose the much more efficient GBSW${}_{+}$. At the same time, we omit the constant weight β=0.025, thus reducing the formula for the predicted value to a simple linear combination: (13)$${\hat{x}}_{n}=b_{1}\cdot {\hat{x}}_{GBSW_{+}}+b_{2}\cdot {\hat{x}}_{GAP_{+}}+\sum_{j=3}^{r}b_{j}\cdot P\left(j-2\right).$$

Given the possibility of using a hardware implementation of the decoder with low power consumption and hardware resources, it is worth noting that this formula is feasible using fixed-point computing. The decoding process requires access to only three image rows (the currently decoded and two previous rows) in the case of r=14 (we use the 12 closest pixels of P(j)) and up to four image rows at r=24.

Since the proposed codec assumes the presence of time asymmetry between the encoder and decoder, a flexible approach is possible in determining the prediction order and in how to choose the prediction coefficients, which are placed in the file header. One approach may be to use ISA, but in most cases, the encoding time may not be acceptable. It is also possible to use the determination of prediction coefficients with the fast classic MMSE method or Algorithm 2 described in Section 3.2, which offers some compromise between the speed of operation and the bit-average level $L_{avg}$.

Unchanged from Multi-ctx is the context-dependent correction of the cumulative prediction error (module CDCCR—Context-Dependent Constant Component Removing—in Figure 5 showing the block coding scheme proposed in this work). For a detailed description of CDCCR, see paper [28]. Similar solutions were previously used in codecs, such as JPEG-LS and CALIC. Removing the constant $C_{mix}$ component associated with one of the 1024 contexts requires a slight modification of Formula (Equation 1) to the following form: (14)$$e_{n}=x_{n}-\left[{\hat{x}}_{n}+C_{mix}\right].$$

Another improvement over Multi-ctx is using a newer prediction error encoder, a combination of an adaptive variation of the Golomb code and a binary arithmetic encoder (see paper [25] for details).

The use of these modifications, despite the lack of code optimizations (such as the introduction of fixed-point operations in place of floating-point operations or the conversion of the code from C++ to assembler), made it possible to reduce the decoding time of a Lennagrey image (with dimensions of 512 × 512 pixels) using an i5 3.4 GHz processor from 0.35 s to 0.245 s (with r=24) for a Multi-ctx decoder. This represents a 30% reduction in time. At the same time, the bit average decreased, even when the prediction order in the encoder was reduced to r=14. The fast classical MMSE approach was used to determine the prediction coefficients instead of ISA or Algorithm 2. A detailed comparison of bit averages with other codecs known from the literature will be discussed in Section 4.

### 3.2. Algorithm for Fast Determination of Prediction Coefficients Using the Non-MMSE Method

As discussed in Section 2.2, Algorithm ISA is characterized by a too-long coding time at high prediction orders. In the literature on determining prediction coefficients using non-MMSE methods, one also encounters attempts to use, for example, genetic algorithms. However, for instance, in works such as [46,47,48], such algorithms were designed for low-order prediction models (r=3 and r=4). Our experiments indicate that it is more difficult with a high prediction order (r>10) to obtain a significantly faster convergence of linear prediction model construction in this way than is the case when using ISA. Thus, there was a need to design a compromising solution that could determine linear predictor coefficients quickly, allowing lower entropy than the classic MMSE approach with forward adaptation. To this end, to select the final prediction model, it was necessary to take advantage of the possibilities offered by methods with backward transformation, which can iteratively improve compression efficiency.

The most common prediction coefficient adaptation methods use least mean square (LMS). Regarding image coding, it is among the simplest but, at the same time, the least efficient methods, even in the normalized version of LMS (NLMS). This is due to two reasons. First, a problematic issue is the selection of the right learning rate μ with different levels of data input variability. In the few works using LMS in lossless image compression, for this reason, the value of μ is set relatively low, such as $2^{-23}$–$2^{-22}$[49], and this results in slow convergence to the expected results. Second, most of the NLMS improvement methods known from the literature, including the selection of a locally variable μ value, are optimized for time series in which we have a one-dimensional input string. This also applies to more computationally complex methods such as RLS.

In contrast, images offer correlations between adjacent pixels in two dimensions. One can largely overcome these difficulties by introducing transformations that use, for example, differential data of neighboring pixel values [50]. This reduces the input signal’s dynamics while increasing the predictive model’s learning speed. Various input data transformations are often used in the literature to achieve faster convergence of adaptive methods [51]. For this purpose, a vector is introduced: (15)A=T·X,
where T is the transformation matrix, the vector $X={\left[P\left(1\right),P\left(2\right),\ldots ,P\left(r\right)\right]}^{T}$ contains the neighborhood pixels of the currently encoded pixel $x_{n}$. The transformed data vector Y is used to calculate the predicted value based on the prediction model included in vector A: (16)$${\hat{x}}_{n}=A\cdot Y.$$

Typically, the matrix T is a square matrix of size r×r, where *r* is the most commonly defined power of two; for example, the Walsh–Hadamard matrix at $r=2^{2}$ looks like this:(17)$$T=\left[\begin{array}{cccc} 1 & 1 & 1 & 1\\ 1 & -1 & 1 & -1\\ 1 & 1 & -1 & -1\\ 1 & -1 & -1 & 1 \end{array}\right]$$

In this case (as in DCT), the first element of vector Y is proportional to the arithmetic mean of vector X, and the sum of the values in the other rows of matrix T is zero. In the case of lossless image coding, this first row can be dispensed with, and at the same time, it can be noted that matrix T does not have to be square. Indeed, there are more rows composed of, for example, the numbers $\left\{-1,0,1\right\}$, in which the sum is zero. The following formula determines their number:(18)∑k=0⌊r2⌋rk·r−kk=∑k=0⌊r2⌋r!k!·k!·(r−2k)!,
where the parameter *k* determines in such a matrix T the number of pairs of numbers $\left\{-1,1\right\}$ and (r−2k) is the number of zeros in a row of the matrix. For example, with r=4, we have 19 rows. Omitting the row of the matrix T consisting of only zeros, we obtain the following matrix: (19)$$T^{T}=\left[\begin{array}{cccccccccccccccccc} -1 & -1 & -1 & -1 & -1 & -1 & 0 & 0 & 0 & 0 & 0 & 0 & 1 & 1 & 1 & 1 & 1 & 1\\ -1 & 0 & 0 & 1 & 1 & 1 & -1 & -1 & 0 & 0 & 1 & 1 & -1 & -1 & -1 & 0 & 0 & 1\\ 1 & 0 & 1 & -1 & 0 & 1 & 0 & 1 & -1 & 1 & -1 & 0 & -1 & 0 & 1 & -1 & 0 & -1\\ 1 & 1 & 0 & 1 & 0 & -1 & 1 & 0 & 1 & -1 & 0 & -1 & 1 & 0 & -1 & 0 & -1 & -1 \end{array}\right],$$
in which a certain symmetry can be observed, and the other half can thus be rejected for this reason. Then, after using Formula (Equation 15), we obtain a vector $Y={\left[y_{1},y_{2},\ldots ,y_{9}\right]}^{T}$ of the data after transformation with nine elements (in general with $r_{A}$ elements) and, consequently, also a vector of nine prediction coefficients $A=[a_{1},a_{2},\ldots ,a_{9}]$. By using the relationships between the data in the Y vector and the data in the X vector, there is a chance of faster convergence by the adaptive LMS-type methods. At the same time, it should be noted that after determining the final prediction model A, it is possible to obtain a prediction model B of order *r* using an inverse transformation, as will be shown later in this section with a practical example. Due to the rapidly increasing number of rows of matrix T along with the prediction row, in the practical implementation, we decided to use only the rows of matrix T in which there is only one pair of non-zero values $\left\{-1,1\right\}$. This choice boils down to the fact that elements $y_{j}$ consist of differences between two pixels. In practice, we reduce this even further to pixels that are immediately adjacent to each other (and available in the decoding procedure); for example, the nearest neighbors of pixel P(1) are P(2), P(3), P(5), and P(7). Then, through experiments, we select (as in work [50]) a subset of these differences that work well with the prediction coefficient adaptation method. Taking into account the prediction model proposed in Formula (Equation 13), we obtain an input data vector of the form $X={\left[{\hat{x}}_{GBSW_{+}},{\hat{x}}_{GAP_{+}},P\left(1\right),P\left(2\right),\ldots ,P\left(r-2\right)\right]}^{T}$. Assuming that we are initializing the vector A=[0,0,…,0], modify Formula (Equation 16) to the form: (20)$${\hat{x}}_{n}=P\left(1\right)+A\cdot Y.$$

We designed two sets of transformations for orders r=14
$\left(r_{A}=27\right)$ and r=24
$\left(r_{A}=43\right)$, respectively. Table 4 shows the experimentally selected values of the Y vector for r=14
(j≤27) and for r=24
(j≤43).

In designing Algorithm 2, we used a modified form of the activity-level classification model (ALCM${}_{+}$), which is closer to MAE minimizing than to MSE, as is the case with NLMS, as a means of adaptively determining the prediction coefficients of A. In the original, ALCM operated on fifth- or sixth-order linear prediction models [52]. After encoding the next pixel, only a select two of these several coefficients were adapted (by a constant value of μ=1/256). In the solution proposed here (see Algorithm 2), as many as eight of the $r_{A}$ of the $a_{j}$ coefficients of vector A are adapted for each successively encoded pixel. For this purpose, four of the lowest and highest values from the Y vector are determined. We denote them as the four smallest values, $P\left(q_{1}\right)\leq P\left(q_{2}\right)\leq P\left(q_{3}\right)\leq P\left(q_{4}\right)$, and the four largest values, $P\left(p_{4}\right)\leq P\left(p_{3}\right)\leq P\left(p_{2}\right)\leq P\left(p_{1}\right)$, respectively. Then, assuming $P\left(q_{1}\right)<P\left(p_{2}\right)$, the adaptation presented in Algorithm 2 is performed.
**Algorithm 2** Prediction coefficients adaptation1: **for all** Pixel $x_{n}$ **do** calculate the estimated value based on Formula (Equation 20)2:  **if** ${\hat{x}}_{n}<x_{n}$ **then**3:   $a_{p_{k}}\left(n+1\right)\leftarrow a_{p_{k}}\left(n\right)+\mu_{k},$ for $k=\left\{1,2,3,4\right\}$4:   $a_{q_{k}}\left(n+1\right)\leftarrow a_{q_{k}}\left(n\right)-\mu_{k},$ for $k=\left\{1,2,3,4\right\}$5:  **else if** ${\hat{x}}_{n}>x_{n}$ **then**6:   $a_{p_{k}}\left(n+1\right)\leftarrow a_{p_{k}}\left(n\right)-\mu_{k},$ for $k=\left\{1,2,3,4\right\}$7:   $a_{q_{k}}\left(n+1\right)\leftarrow a_{q_{k}}\left(n\right)+\mu_{k},$ for $k=\left\{1,2,3,4\right\}$8:  **end if**9: **end if**

Learning coefficients are determined as follows: (21)$$\mu_{k}=\alpha_{k}\cdot \max\left\{2^{-\gamma (t)};\mu_{0}\right\},$$
where
(22)$$\mu_{0}=\frac{\left|e_{n}\right|}{\delta \left(t\right)\cdot \sum_{k=1}^{4}\alpha_{k}\cdot \left(P\left(p_{k}\right)-P\left(q_{k}\right)\right)},$$
where $\alpha_{k}=\left\{1,0.75,0.5,0.5\right\}$, and the values of γ(t) and δ(t) change in successive iterations. This is because the entire image is scanned five times, and in each successive iteration, the influence of the learning rate should be reduced (according to the principle from coarse to fine-tuning). Experimentally selected parameters are $\gamma \left(t\right)=\left\{10,12,14,15,17\right\}$, $\delta \left(t\right)=\left\{250,1200,5000,30000,100000\right\}$, respectively, for $t=\left\{1,2,\ldots ,5\right\}$. Thus, we obtain (regardless of the prediction order) the final form of vector A after only five initial image encodings, which we transform into vector B. This requires Table 5 for r=14 or Table 6 for r=24, which contains a set of transformations of coefficients from the domain of vector A to vector B. Thus, after the adaptation stage, we can use the target Formula (Equation 13) for the predicted value in the encoder and in the decoder. The final step is to reduce, using Formula (Equation 3), the accuracy of the prediction coefficients’ notation to N=12 bits after the decimal point, after which we perform the full final encoding according to the scheme shown in Figure 5. In the case of the Algorithm 2 implementation, the total time to determine the prediction coefficients and encode the Lennagrey image is 1.515 s with r=14 and 1.925 s with r=24 (decoding times are 0.24 and 0.245 s, respectively).

### 3.3. Features of the Proposed Solution

In Figure 5, we have shown a block diagram of the encoder proposed here (with four separate stages), in which the data processing process proceeds as follows (Algorithm 3):
**Algorithm 3** Steps of coder data processing1: For each successively encoded pixel $x_{n}$:2: Calculate the predicted value (Formula (Equation 13)—stage 1) and the prediction error $e_{n}$ after considering the context-dependent component of the constant $C_{mix}$ (Formula (Equation 14)—stage 2);3: Convert the prediction error $e_{n}$ to a bit stream using an adaptive Golomb encoder (stage 3);4: Encode the bit stream from stage 3 using an adaptive binary arithmetic encoder;5: Return to step 2 if there are still pixels to code.

The decoding process, in turn, is described by Algorithm 4.
**Algorithm 4** Steps of decoder data processing1: For each successively encoded pixel $x_{n}$:2: Decode the bit sequence from the input using an adaptive binary arithmetic encoder, obtaining the Golomb code word and prediction error $e_{n}$ after considering the context-dependent component of the constant $C_{mix}$ (Formula (Equation 14)—stage 2);3: Convert the Golomb word to a prediction error value $e_{n}$;4: Calculate the predicted value (Formula (Equation 13)) and $C_{mix}$, and then add these values to $e_{n}$, obtaining the decoded pixel value $x_{n}$;5: Return to step 2 if there are still pixels to decode.

In this work, we focus on analyzing the details of stage 1 (stages 2–4 are discussed in the paper [25,28]). Section 2.2 showed that non-MMSE minimization in a simple predictive model achieves higher compression efficiency than the classical approach. The coding and decoding system was designed to allow decisions on which approach to determining prediction coefficients to use (MMSE, ISA, or type ALCM${}_{+}$ adaptation). As a compromise solution in the final version of the encoder, we proposed a predictor of order r=24 and the calculation of prediction coefficients in an adaptive manner (ALCM${}_{+}$), where, in five iterations of the initial encoding, an online predictive model is tuned (see Algorithm 2). It should be noted here that there is no need to use entropy coding stages at the initial stage (stages 3 and 4 in Figure 5). In addition, the time periods for calculating prediction coefficients (in the ALCM${}_{+}$ version) and encoding and decoding times are linearly dependent on the number of image pixels.

## 4. Summary

### 4.1. Experimental Results

Table 7 compares bit averages (for two sets of standard test images, the first consisting of 9 images and the second of 45) obtained by the proposed method using three different approaches to determining prediction coefficients. The first set is the most commonly used in papers on lossless image compression (Figure 6). The second set includes the first set—the whole dataset is available at [53]. The final compression method proposed in this work was determined to be the one in which Algorithm 2 was used to determine the prediction coefficients with a prediction order of r=24.

Table 8 compares the bit averages (for nine standard test images) of the proposed solution with those obtained by several methods with fast encoding and decoding times known from the literature. The proposed solution offers a 7.9% lower bit average relative to the popular JPEG-LS codec. It also presents the results of the high-performance LA-OLS method [25], characterized by symmetric, significantly longer encoding and decoding times than the other solutions. Although the bit average for the LA-OLS method is lower than when using the technique presented in this work (using Algorithm 2), the Lennagrey image encoding time of 5.86 s (at 512 × 512 pixel resolution using i5 3.4 GHz processor) using LA-OLS is more than three times longer. The decoding time is as much as 23.9 times longer. In addition, compared to the relatively fast Multi-ctx method, the solution proposed here offers a 30% shorter decoding time (amounting to 0.245 s with r=24) while having a lower bit average. The decoding time of the solutions compared here depends linearly on the number of pixels of the image being decoded, reflecting the implementation complexity of the decoding process. These results refer to a non-optimized implementation in C++ (without parallelization and use of GPU resources). Therefore, decoding time, although shorter than LA-OLS and Multi-ctx, is inferior to fully optimized solutions (such as CALIC and JPEG-LS), where decoding time is measured in milliseconds. It should be noted here that the decoding algorithm, due to its simplicity, lends itself to easy hardware implementation.

### 4.2. Conclusions

The analysis of solutions for lossless image coding presented in the introduction leads to the conclusion that the choice of compression method depends on user preferences. Based on the assumption that we usually encode an image once and decode it multiple times, we should care about a relatively fast decoding process. For this purpose, one should use time-asymmetric predictive methods with forward adaptation with adjustable implementation complexity in the encoding process. To achieve high compression efficiency, use linear or nonlinear prediction at the data modeling stage and, for example, arithmetic coding of prediction errors.

In this work, a method for the lossless compression of images with a fast decoding time and flexible selection of encoder parameters is presented, proposing, in addition to the classical method of determining prediction coefficients based on minimizing the mean square error, two other algorithms of the non-MMSE type. The proposed authors’ algorithm (based on ALCM${}_{+}$ type adaptation) for determining prediction coefficients by the non-MMSE method is characterized by a complexity linearly dependent on the number of pixels of the encoded image (unlike, for example, the simplex method, where in extreme cases exponential complexity can be encountered). In the proposed solution, the data modeling stage was based on linear and nonlinear prediction, and a simple block for removing the context-dependent constant component was used. The prediction errors produced after applying the prediction are subjected to two-stage compression using an adaptive Golomb code and a binary arithmetic code.

## Figures and Tables

**Figure 1 entropy-25-00156-f001:**
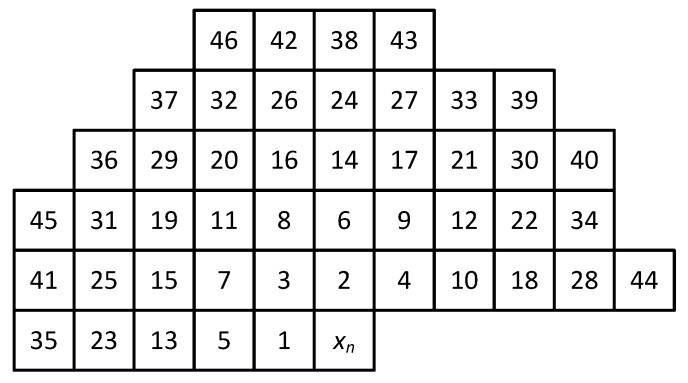
Neighborhood pixel numbering.

**Figure 2 entropy-25-00156-f002:**
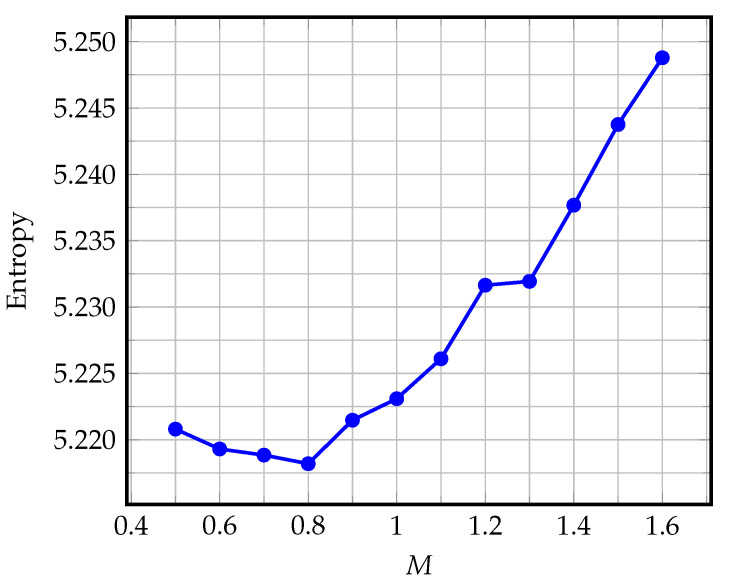
Entropy as a function of the parameter *M* for the Noisesquare image encoded with the linear predictor of the order r=8.

**Figure 3 entropy-25-00156-f003:**
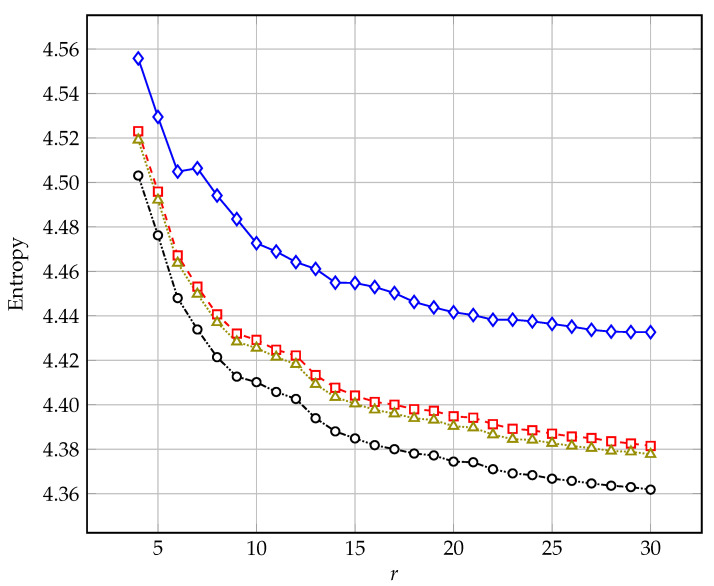
Average entropy value (for the set of 45 test images) as a function of the prediction order obtained using four minimization criteria: MMSE (M=2)—marked with diamonds; MMAE (M=1)—marked with squares; Minkowski (M=0.75)—marked with triangles; entropy function *H*—marked with circles.

**Figure 4 entropy-25-00156-f004:**
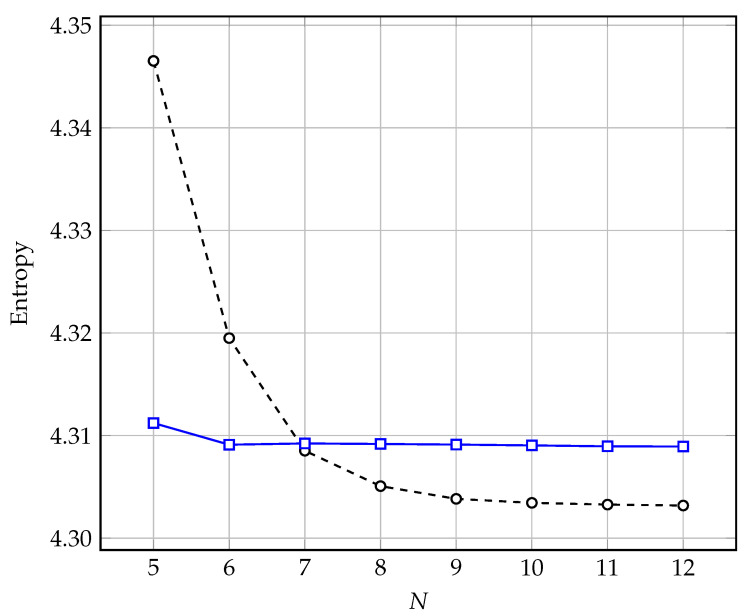
The average value of entropy as a function of *N* fractional bits of precision of the prediction coefficients (for the set of 45 images at r=14) obtained by two methods: classic MMSE—dashed line, ISA based on Formula (Equation 6) at M=2—solid line.

**Figure 5 entropy-25-00156-f005:**
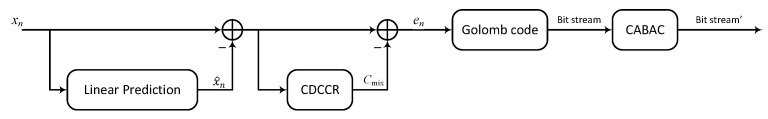
Block diagram of the proposed cascade coding.

**Figure 6 entropy-25-00156-f006:**
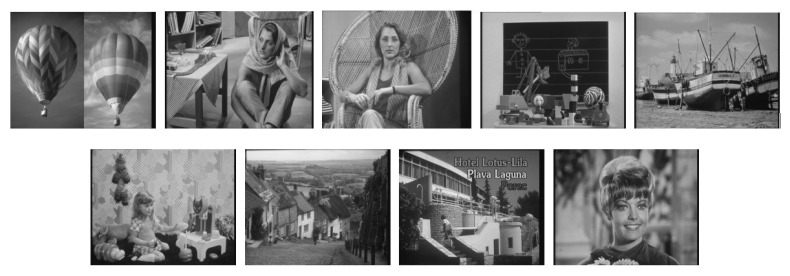
A base test set of nine pictures (Ballon, Barb, Barb2, Board, Boats, Girl, Gold, Hotel, Zelda).

**Table 1 entropy-25-00156-t001:** Number of tests in brute force method with N=5, r≤5.

	r=2	r=3	r=4	r=5
$N_{BF}$	$2^{7}$	$2^{14}$	$2^{21}$	$2^{28}$
Reduced number of tests	95	11,135	1,265,346	148,311,811
Percent reduced number of tests relative to $N_{BF}$	74.219	67.963	60.336	55.251

**Table 2 entropy-25-00156-t002:** The set of prediction coefficients corresponding to each context of the method GAP${}_{+}$.

Coeff. \*k*	1	2	3	4	5	6	7
$b_{1}$	1/2	7/8	5/4	3/8	1/4	2	0
$b_{2}$	1/2	3/8	1/4	7/8	5/4	0	2
$b_{3}$	−1/4	−3/16	−1/8	−3/16	−1/8	0	0
$b_{4}$	1/4	3/16	1/8	3/16	1/8	0	0
$b_{5}$	0	−1/4	−1/2	0	0	−1	0
$b_{6}$	0	0	0	−1/4	−1/2	0	−1

**Table 3 entropy-25-00156-t003:** Comparison of entropy values of different prediction methods.

Image	MED	MED${}_{+}$	GAP${}_{+}$	FBLP${}_{+}$ [43]	SOLP [44]	GBSW${}_{+}$	Blend-7 [45]	Multi-ctx
Balloon	3.120	3.111	2.997	3.019	2.991	3.023	2.93	2.861
Barb	5.204	5.201	5.009	4.918	5.022	4.942	4.90	4.793
Barb2	5.181	5.170	4.998	5.050	4.976	5.034	5.02	4.910
Board	3.947	3.935	3.892	3.911	3.794	3.768	3.81	3.732
Boats	4.307	4.295	4.229	4.306	4.220	4.227	4.16	4.125
Girl	4.207	4.189	4.044	4.086	4.036	3.961	3.91	3.891
Gold	4.716	4.715	4.699	4.719	4.699	4.785	4.65	4.609
Hotel	4.732	4.727	4.672	4.670	4.624	4.582	4.58	4.519
Zelda	4.113	4.108	3.936	4.024	3.938	3.930	3.90	3.905
Average	4.392	4.383	4.275	4.300	4.256	4.250	4.207	4.149

**Table 4 entropy-25-00156-t004:** A set of transformations of vector X elements into vector Y.

*j*	$y_{j}$	*j*	$y_{j}$	*j*	$y_{j}$
1	P(1)−P(2)	16	P(10)−P(4)	31	P(9)−P(17)
2	P(1)−P(3)	17	P(7)−P(11)	32	P(14)−P(16)
3	P(2)−P(3)	18	P(4)−P(12)	33	P(10)−P(18)
4	P(1)−P(5)	19	P(4)−P(9)	34	P(15)−P(19)
5	P(2)−P(6)	20	${\hat{x}}_{GAP_{+}}-P\left(4\right)$	35	P(11)−P(20)
6	$P\left(1\right)-{\hat{x}}_{GBSW_{+}}$	21	P(6)−P(8)	36	P(12)−P(22)
7	P(4)−P(2)	22	P(6)−P(9)	37	P(11)−P(15)
8	P(3)−P(6)	23	P(10)−P(9)	38	P(8)−P(16)
9	P(4)−P(6)	24	P(10)−P(12)	39	P(7)−P(15)
10	P(1)−P(7)	25	P(9)−P(12)	40	P(7)−P(13)
11	$P\left(2\right)-{\hat{x}}_{GBSW_{+}}$	26	P(8)−P(11)	41	P(8)−P(20)
12	P(3)−P(7)	27	$P\left(4\right)-{\hat{x}}_{GBSW_{+}}$	42	P(11)−P(19)
13	${\hat{x}}_{GAP_{+}}-P\left(1\right)$	28	P(5)−P(13)	43	P(12)−P(21)
14	P(2)−P(8)	29	P(6)−P(14)		
15	P(2)−P(9)	30	P(13)−P(15)		

**Table 5 entropy-25-00156-t005:** A set of transformations of coefficients from the domain of vector A to the vector B at r=14.

*j*	$b_{j}$	*j*	$b_{j}$
1	$-a_{6}-a_{11}-a_{27}$	8	$-a_{5}-a_{8}-a_{9}+a_{21}+a_{22}$
2	$a_{13}+a_{20}$	9	$-a_{10}-a_{12}+a_{17}$
3	$1+a_{1}+a_{2}+a_{4}+a_{6}+a_{10}-a_{13}$	10	$-a_{14}-a_{21}+a_{26}$
4	$-a_{1}+a_{3}+a_{5}-a_{7}+a_{11}+a_{14}+a_{15}$	11	$-a_{15}-a_{19}-a_{22}-a_{23}+a_{25}$
5	$-a_{2}-a_{3}+a_{8}+a_{12}$	12	$a_{16}+a_{23}+a_{24}$
6	$a_{7}+a_{9}-a_{16}+a_{18}+a_{19}-a_{20}+a_{27}$	13	$-a_{17}-a_{26}$
7	$-a_{4}$	14	$-a_{18}-a_{24}-a_{25}$

**Table 6 entropy-25-00156-t006:** A set of transformations of coefficients from the domain of vector A to the vector B at r=24.

*j*	$b_{j}$	*j*	$b_{j}$
1	$-a_{6}-a_{11}-a_{27}$	13	$-a_{17}-a_{26}+a_{35}+a_{37}+a_{42}$
2	$a_{13}+a_{20}$	14	$-a_{18}-a_{24}-a_{25}+a_{36}+a_{43}$
3	$1+a_{1}+a_{2}+a_{4}+a_{6}+a_{10}-a_{13}$	15	$-a_{28}+a_{30}-a_{40}$
4	$-a_{1}+a_{3}+a_{5}-a_{7}+a_{11}+a_{14}+a_{15}$	16	$-a_{29}+a_{32}$
5	$-a_{2}-a_{3}+a_{8}+a_{12}$	17	$-a_{30}+a_{34}-a_{37}-a_{39}$
6	$a_{7}+a_{9}-a_{16}+a_{18}+a_{19}-a_{20}+a_{27}$	18	$-a_{32}-a_{38}$
7	$-a_{4}+a_{28}$	19	$-a_{31}$
8	$-a_{5}-a_{8}-a_{9}+a_{21}+a_{22}+a_{29}$	20	$-a_{33}$
9	$-a_{10}-a_{12}+a_{17}+a_{39}+a_{40}$	21	$-a_{34}-a_{42}$
10	$-a_{14}-a_{21}+a_{26}+a_{38}+a_{41}$	22	$-a_{35}-a_{41}$
11	$-a_{15}-a_{19}-a_{22}-a_{23}+a_{25}+a_{31}$	23	$-a_{43}$
12	$a_{16}+a_{23}+a_{24}+a_{33}$	24	$-a_{36}$

**Table 7 entropy-25-00156-t007:** Comparison of bit averages (for two sets of standard test images) obtained by the proposed method with different algorithms for determining prediction coefficients.

Image	MMSE r=14	Algorithm 2 r=14	ISA r=14	MMSE r=24	Algorithm 2 r=24	ISA r=24
Balloon	2.742	2.748	2.732	2.733	2.733	2.720
Barb	4.238	4.238	4.229	4.221	4.211	4.196
Barb2	4.416	4.421	4.411	4.406	4.401	4.393
Board	3.446	3.449	3.436	3.436	3.425	3.417
Boats	3.717	3.711	3.703	3.707	3.696	3.688
Girl	3.612	3.624	3.604	3.609	3.612	3.595
Gold	4.310	4.304	4.302	4.299	4.293	4.290
Hotel	4.180	4.176	4.158	4.179	4.169	4.153
Zelda	3.634	3.620	3.619	3.621	3.598	3.603
Average (9 images)	3.810	3.810	3.799	3.801	3.793	3.784
Average (45 images)	4.045	4.038	4.029	4.032	4.021	4.010

**Table 8 entropy-25-00156-t008:** Comparison of bit averages (for nine standard test images) obtained by several methods known from the literature.

Image	JPEG-LS	CALIC	Blend-7 [45]	HBB [54]	Multi-ctx	Proposed Solution	LA-OLS
Balloon	2.889	2.78	2.84	2.80	2.727	2.733	2.576
Barb	4.690	4.31	4.43	4.28	4.243	4.211	3.832
Barb2	4.684	4.46	4.57	4.48	4.421	4.401	4.214
Board	3.674	3.51	3.57	3.54	3.467	3.425	3.288
Boats	3.930	3.78	3.84	3.80	3.730	3.696	3.537
Girl	3.922	3.72	3.76	3.74	3.664	3.612	3.467
Gold	4.475	4.35	4.42	4.37	4.310	4.293	4.198
Hotel	4.378	4.18	4.29	4.27	4.171	4.169	4.040
Zelda	3.884	3.69	3.79	3.72	3.700	3.598	3.499
Average	4.058	3.864	3.946	3.889	3.826	3.793	3.628

## Data Availability

The datasets for this study are available upon request from the corresponding author.

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
