# Peer review of "Lossless Image Coding Using Non-MMSE Algorithms to Calculate Linear Prediction Coefficients"

_entropy, 2023, doi:10.3390/e25010156_

Round 1

Reviewer 1 Report

    In this manuscript, the authors proposed a lossless image compression method using non-MMSE algorithm to calculate linear prediction coefficients. The experimental results show that the proposed method achieves a lower bit average compared to the mean-square error and the proprietary algorithms. The related methods and the proposed method are well written. However, the organization of the manuscript should be improved, and some points should be modified:

1. Section “Summary” should be divided into two sections, such as “Experimental results” and “Conclusions”.

2. The appearance of the test images used in the experiments should be presented, and the source should be clarified.

3. It would be better to test the performances on some larger datasets, such as Bossbase, etc.

4. The result of the decoding time is insufficient. It would be better to evaluate the time complexity among different methods on the test images or a dataset.

Reviewer 2 Report

The article's authors did a good job of presenting a lot of information, but there are many gaps that need to be addressed in order to improve it. The article's drawbacks are listed below.

1. A brief summary of the method, the percentage improvement in decoding time, and the percentage improvement in lower bit average when compared to competing algorithms should all be included in the abstract.

2. The computational complexity of the approach is increased by the two stage compression process. So, include a big-O notation analysis of computational complexity.

3. The statements in lines 45 and 46 must be justified by the authors. Otherwise, authors must include a suitable citation to support that claim.

4. In the article's "Introduction section," authors should add a section organisation.

5. The article's literature review section contains insufficient descriptions of the literature.

6. There is no need for a detailed explanation of how linear and nonlinear prediction are applied in lossless image coding in the proposed work. Therefore, condense Section 2 of the article.

7. The entire algorithm is not adequately conveyed by a block diagram. Therefore, it is necessary to create a block diagram that accurately depicts the entire method.

8. The primary evaluation criterion for the article, decoding time comparisons with competing algorithms, has not been done by the authors.

9. In order for the method to be well understood, authors should describe it step by step.

10. Not the "Summary" section, but the "Conclusion" section should be written in the article.

11. Only a few recent articles—roughly two—are added to the reference list. Therefore, some recent articles from 2021 and 2022 must be added.

Reviewer 4 Report

The paper addresses an important topic and the proposed method is interesting.  However, the readability of the paper should be improved, and for this, details are to be provided in a number of points.

1 Abstract

It is too short.  Such a brief description does not help to summarize the proposed method and achieved results.

2 Introduction

It looks a bit confusing.  What is the main claim? How do you evaluate method performance and which are the comparisons to be made?  Futhermore, we miss a better connection between the Introduction and Section 2, which treats the baseline developments for the proposed method.  Why should these developments be the substrate of your proposal, why do they fit the present needs in the target application?

3 Figure 2: what is the intended message in it, how the figure helps understanding your rationale? 

Likewise, Figure 3 deserves clarification: "... Figure 3 shows the comparison of average entropy values obtained using the ISA algorithm (for a base of 45 test images)...".  It is not also clear whether this figure is general enough or is just a motivation for the proposed method.  Where did those 45 test images come from? Please, also elaborate further on the reason for using M=0.6 in Equation 7.

4 From the text: "... The number of modifying vectors is NΔB = r · (r − 1)...".  Please, clarify this statement.

5 Please, make your figure captions clear: 

Figure 4. The average value of entropy as a function of N fractional bits precision of the prediction coefficients (for the set of 45 images at r = 14) obtained by two methods.

Figure 5: the caption  (" Block diagram of the proposed cascade coding.") does not help much in understanding.  From the text: "... Unchanged from Multi-ctx is the use of context-dependent correction of the cumulative prediction error (module CDCCR - Context-Dependent Constant Component Removing in Figure 5 showing the block coding scheme proposed in this work) - for a detailed description of CDCCR, see paper [41]...".  You bring too much information from previous works with regard to the writing of the present paper, causing the readability to be impaired, as the necessary information for an adequate evaluation of the proposed method is often missing.

6 How did you come to the set of transformation displayed in Table 4?

Grammatical errors were detected.

Round 2

Reviewer 1 Report

The authors have addressed all the comments, and I recommend the acceptance.

Author Response

The authors thank the reviewer for his valuable comments, which helped improve the article’s final form.

Reviewer 3 Report

Some editing of English language needs to be improved.

Author Response

Authors thank the reviewer for his valuable comments, which helped improve the article’s final form.
Authors would like to inform the reviewer that the article has been checked for language correctness by the publisher's service.

Reviewer 4 Report

The new version was found to improve upon the previous one.  However, few grammatical errors were still detected and some sentences are still confusing.  Despite understanding the authors' arguments on how difficult is to balance the description of previous works and the present proposal, I still find it quite difficult to follow the paper, as a lot of material has to be consulted to understand what is being proposed.  In the end, the highlights of the proposed method get dimmed in the general sense. 

Please, refer to Algorithm 3 and Algorithm 4 in the text.

Author Response

The main authors' assumption is reflected in the paper’s title. It proposes an author's method for determining prediction coefficients by a non-MMSE method with a complexity linearly dependent on the number of pixels of the encoded image (in contrast, for example, to the simplex method, where exponential complexity can be encountered in extreme cases). The key idea is contained in Section 3.2, and the practical application is indicated in the overall encoding/decoding process (Section 3.3).

The sentence was added in Section 4.2: "The proposed authors' algorithm (based on ALCM+ type adaptation) for determining prediction coefficients by the non-MMSE method is characterized by a complexity linearly dependent on the number of pixels of the encoded image (unlike, for example, the simplex method, where in extreme cases exponential complexity can be encountered)."

Authors would like to inform the reviewer that the article has been checked for language correctness by the publisher's service.
